# Cathodic Protection of Mild Steel Using Aluminium-Based Alloys

**DOI:** 10.3390/ma15041301

**Published:** 2022-02-10

**Authors:** Maria del Rosario Silva Campos, Carsten Blawert, Nico Scharnagl, Michael Störmer, Mikhail L. Zheludkevich

**Affiliations:** 1Functional Surfaces, Institute of Surface Science, Helmholtz-Zentrum Hereon GmbH, Max-Planck-Straße 1, 21502 Geesthacht, Germany; carsten.blawert@hereon.de (C.B.); nico.scharnagl@hereon.de (N.S.); mikhail.zheludkevich@hereon.de (M.L.Z.); 2Multifunctional Interfaces, Institute of Photoelectrochemistry, Helmholtz-Zentrum Hereon GmbH, Max-Planck-Straße 1, 21502 Geesthacht, Germany; michael.stoermer@hereon.de; 3Faculty of Engineering, Institute for Materials Science, University of Kiel, Kaiserstraße 2, 24143 Kiel, Germany

**Keywords:** galvanic corrosion, steel, cathodic protection, aluminium alloys

## Abstract

Typically, steel is protected from corrosion by employing sacrificial anodes or coatings based on Zn, Mg, Al or Cd. However, stricter environmental regulations require new environmentally friendly alternatives to replace Cd. Traditionally, Al-based anodes have been employed to cathodically protect steel in marine applications or as ion vapour deposition (IVD)-Al sacrificial coatings for aerospace applications. However, Al tends to passivate, thus losing its protective effect. Therefore, it is important to identify possible alloys that can provide a constantly sufficient current. In this study, Al-X alloys (X = Ag, Bi, Ca, Cr, Cu, Ga, Gd, In, Mg, Mn, Ni, Sb, Si, Sn, V, Ti, Zn and Zr) were firstly tested for a screening of the sacrificial properties of binary systems. Al-0.5Cr, Al-1Sn, Al-0.2Ga, Al-0.1In, Al-2Si and Al-5Zn alloys were suggested as promising sacrificial Al-based alloys. Suitable heat treatments for each system were implemented to reduce the influence of the secondary phases on the corrosion properties by minimising localised attack. extensive evaluation of the corrosion properties, including galvanic coupling of these alloys to steel, was performed in the NaCl electrolyte. A comparative analysis was conducted in order to choose the most promising alloy(s) for avoiding the passivation of Al and for efficient cathodic protection to steel.

## 1. Introduction

High-strength steel is extensively used for structural applications in a diverse range of industries, such as energy, construction, oil and gas, transportation, etc. However, steel components are susceptible to corrosion due to direct exposure to aggressive environmental conditions, such as marine atmospheres. Steel components are therefore mostly protected with sacrificial coatings to prevent this harmful process [1]. Particularly, cadmium has been used as a sacrificial metallic coating for decades due to its excellent resistance to chloride-containing environments in the aerospace industry [2]. Electroplated cadmium coatings are usually achieved by using cyanide or sulphate solutions. Unfortunately, during electroplating, part of the hydrogen diffuses into the steel, resulting in hydrogen embrittlement. [3,4]. Despite the fact that the absorption of the hydrogen by the steel represents a potential problem, this absorbed hydrogen can be completely released by a heating process in a furnace, leaving the mechanical properties intact [3,5]. Moreover, adverse environmental and health effects are now associated with the use of cadmium and its compounds for industrial engineering applications. Thus, the replacement of electroplated cadmium worldwide is mandatory [6,7,8].

Alternative sacrificial materials can be used to replace the electroplated cadmium. The properties of the alternative materials should include good galvanic compatibility with other alloys, sacrificial behaviour stability to steels and good barrier properties.

Aluminium-based alloys are extensively used for structural applications in the transportation industry due to their high strength-to-weight ratios [9]. They are also attractive as anodes in the cathodic protection of steels in seawater due to their low cost and high current capacity [10]. The main disadvantage of aluminium as a sacrificial anode is its natural ability to form a very stable and passive oxide layer, which hinders corrosion of the metal itself and shifts the metal potential to less-active values [11]. There are two electrochemical methods for preventing the corrosion of steel in water, particularly seawater via cathodic protection: (1) impressed current cathodic protection, which involves the application of an external current onto the steel in order to bring its potential to within a cathodic range; and (2) sacrificial cathodic protection, which attaches more active metal to steel, protecting it via galvanic corrosion [10,12,13]. The more-active metal or sacrificial anode is deliberately forced to corrode in order to prevent the steel from degradation. In seawater, steel corrodes severely at potentials above −600 mV/SCE; therefore, the potential range necessary to protect steel is typically between −850 mV and −1100 mV/SCE. Other sources propose a higher maximum of −780 mV/SCE [14,15,16]. According to NACE a potential of −900 mV (Ag/AgCl) is required to protect cathodically steel in saline environments [17]. Nowadays, aluminium-based alloys are still capturing attention as environmentally friendly sacrificial anodes [18]. The effectiveness of such an aluminium anode relies on the alloying elements whose main function is to impede the formation of a protective oxide film on the surface, thereby activating the aluminium [11,19]. Moreover, alloying additions are used in order to shift the potential of aluminium towards more negative values and to generate a homogeneous attack [11]. For this study, we considered our preliminary evaluations of several binary Al-X alloys (X = Ag, Bi, Ca, Cr, Cu, Ga, Gd, In, Mg, Mn, Ni, Sb, Si, Sn, V, Ti, Zn and Zr). A general overview of the corrosion properties, including corrosion current density, corrosion potential and passivation range of the binary alloys, is shown in Figure 1. Additionally, a commercial sacrificial anode alloy (Al-3Zn-Ti-Mn-In) and a chromated Cd layer were also measured, only for comparison purposes. It should be noted that these results have been used as a background for the selection of the materials employed in the current study. Therefore, Sn, Cu and Ag could be considered as the main alloying elements that avoid aluminium passivation, while In and Bi could be considered good activators. Moreover, additions of Si and Zn also reduce the tendency of aluminium to passivation, and in the case of Zn in combination with In and Ti, passivation no longer occurs in the classic sacrificial anode alloy with Al-3Zn-Ti-Mn-In. Surprisingly, Mg and Mn seem to be unsuitable because they behave comparably to pure Al. For further evaluations, we selected traditional alloying elements for aluminium anodes such as Zn, In, Sn, Cr, Ga and Si based on features described below.

Additions of zinc weaken the aluminium oxide on the metal’s surface. Therefore, the most efficient way to produce sacrificial aluminium-based alloy anodes is by adding 5 wt.% of zinc, since the formation of Al-Zn phases significantly enhances not only the metallurgical, but also the electrochemical properties of the anodes [19,20]. The presence of indium in Al-based alloy increases the adsorption of chloride ions at the electrode surface, leading to an activation of Al [21,22,23,24]. On the other hand, minor additions of Cr possess moderate corrosion resistance properties [25]. Moshier et al. [26] found that Cr promotes passivity in Al if it can be retained in solid solution without the formation of precipitates. The latter can serve as an active microgalvanic cell. Furthermore, small amounts of Ga are added to aluminium to destabilise the oxide film, generating active corrosion [10,27]. You et al. [28] found that at least 0.1 wt.% Ga is necessary to improve the activity of the Al-based anode. In addition, Sn reduces the ionic resistance of oxide films on Al-Sn alloys [29]. This is possible due to the creation of additional cation vacancies by entering tin as Sn^4+^, which is responsible for improving cathodic protection properties [29]. Higher alloying contents of Si cause a greater likelihood of localised corrosion in Al alloys [30].

One of the main goals of this study is to identify which alloying elements could influence the natural behaviour of Al. In parallel, the alloys’ composition is simplified to the extent that they can be easily applied as sacrificial coatings by the PVD process. Note that the specific coating performance will be discussed in a later publication. The evaluation of the corrosion properties includes open circuit potential; potentiodynamic polarisation; and galvanic coupling between steel and these binary systems during alternating immersion and salt spray tests (SST). Additionally, a commercial sacrificial anode (Al-3Zn-Ti-Mn-In) is included in this study as a reference. A comparative analysis is performed to understand the effect of alloying and microalloying in order to identify the most promising Al-based sacrificial alloy and to design a more efficient sacrificial Al-based coating to substitute Cd-plating.

## 2. Materials and Methods

### 2.1. Materials

The composition of the Al-X alloys was selected based on the solid solubility of the alloying elements in the aluminium matrix [31,32,33,34,35,36] and considering the compositions used in our previous investigations (Figure 1) in order to observe the effect on their corrosion properties and their performance as sacrificial alloys. The binary Al-X alloys were prepared by gravity casting, using either pure alloying elements or master alloys: aluminium (99.996%, Hydro, Bonn, Germany), zinc (>99.99% Carl Roth, Karlsruhe Germany), silicon (99.999%, Alfa Aesar, Kandel, Germany), indium (99,995%, Grirem Advanced Materials Co. Ltd., Sichuan, China), gallium (99.9%, Grirem Advanced Materials Co. Ltd., Sichuan, China), tin 99.99% (Merck, Darmstadt, Germany) and AlCr75 master alloy (AMG Titanium Alloys & Coatings, Chemnitz, Germany). A commercial sacrificial anode alloy (Al-3Zn-Ti-Mn-In) was purchased from Grillo-Werke AG, in Germany. The commercial sacrificial anode was included in this study as a reference. For comparison, pure Al ingots were also prepared using the same casting procedure. The melting and alloying were performed in a graphite crucible with the aid of a stirrer (graphite) without inert gas. After melting, the aluminium was alloyed and the melt was kept for 30 min. Before casting, the melt was stirred for 10 min. Boride-coated steel molds were used for casting, which had been preheated to 350 °C before pouring. The casted alloy rods were 18 mm in diameter and approx. 200 mm in length. Some of the rods were subsequently subjected to a solution treatment at 220 °C, 530 °C and 600 °C (see Table 1) for 24 h and then quenched in cold water (~10 °C) at room temperature to produce a microstructure as free of precipitations as possible. These solution treatment temperatures were selected based on the phase diagrams of the different alloying elements, in order to keep the alloys either in the solid solution region or close as possible to this region [31,32,33,34,35,36]. However, Al-Si system preliminary evaluations showed that additions of 1 wt.%. Si did not reduce the passivation of aluminium (Figure 1); therefore, its concentration was increased to 2 wt.% Si, thus generating more intermetallic phases but reducing the passivation range. For commercial anode Al-3Zn-Ti-Mn-In, the same annealing temperature at 530 °C as Al-5Zn was chosen due to Zn being at its highest concentration in the commercial anode. The chemical compositions of the alloys was determined using X-ray fluorescence spectroscopy (Bruker S8 Tiger, Germany). Chemical composition is listed in Table 2. Weldable heat-treatable steel (1.7734.4) with nominal composition (in wt.%) of 0.165 C, 0.14 Si, 0.85 Mn, 0.008 P, 0.011 S, 1.36 Cr. 0.84 Mo and 0.22 V (CP Autosport GmbH, Büren, Germany) was used to evaluate the anode sacrificial performance of the binary Al-based alloys.

### 2.2. Microstructural Analysis

The as-cast and solution treatment samples were cut from the ingots and were 18 mm in diameter and 10 mm in thickness. Afterwards, the samples were embedded in epoxy resin (Demotec 30, Nidderau, Germany) then wet-ground with SiC paper up to 2500 grit and subsequently polished with a mixture of 1 µm diamond suspension and water-free colloidal silica OPS^TM^ (Struers, Willich, Germany), rinsed with ethanol and dried with air.

The precipitated phases in the alloys were characterised using X-ray diffraction (XRD, Bruker AXS D8 Advance, Karlsruhe, Germany) equipped with Cu Kα radiation. The diffraction patterns were collected at 40 kV and 40 mV using the 2θ scan range from 10° to 90° with a step size of 0.4° and dwell time of 16 s.

The microstructures were analysed by scanning electron microscopy (SEM, Tescan Vega3, Brno, Czech Republic) equipped with energy dispersive X-ray spectroscopy (EDS) in backscattering mode (BSE) using an accelerating voltage of 15 kV and working distance of 15 mm.

### 2.3. Electrochemical Measurements

For the corrosion properties, evaluation open circuit potential, potentiodynamic polarisation and galvanic current measurements were performed. Specimens 18 mm in diameter and 10 mm in thickness were cut from the ingots and wet-ground with SiC paper up to 1200 grit, rinsed with ethanol and dried with forced air. A traditional three-electrode cell was used for the experiments, composed of a Pt mesh as a counter electrode, Ag/AgCl (3M) electrode as reference electrode and sample (with 0.5 cm^2^ exposed area) as working electrode in aerated 3.5 wt.% NaCl electrolyte (~330 mL) and magnetic stirring at a speed of 200 rev min^−1^ at 21.5 ± 0.5 °C. The cell was connected to a potentiostat (Gill AC, from ACM Instruments, Cartmel, UK).

Open-circuit potential was recorded for 2 h, then the polarisation scan was started at –250 mV relative to the free-corrosion potential with a scan rate of 12 mVmin^−1^. The test was terminated when a corrosion current of 4 mA was exceeded. The corrosion current was determined at the intersection point of the cathodic branch with the vertical by the corrosion potential.

In order to evaluate the performance of the Al-based alloys as possible sacrificial coatings without possible contribution of internal galvanic cells, following evaluations were only carried out with the annealed samples:

Galvanic coupling in form of zero-resistance ammeter (ZRA) measurements was conducted in a 330 mL modified cell with two working electrodes, where mild steel and Al-based alloy were placed, respectively. The distance between working electrodes was 7 cm with 0.5 cm^2^-exposed areas and two Ag/AgCl reference electrodes. The galvanic current and mixed potential were recorded every 15 s over periods of 21 h for 21 days.

Alternating immersion and salt spray tests were carried out with the heat-treated samples in 5 wt.% NaCl solution for 6 weeks. The specimens consisted of discs of Al-based alloys (10 mm thickness and 18 mm diameter), provided with a 2 mm-deep and 1 mm-wide notch. The calculated exposed cathodic area was 22 mm^2^ while the anode area was 798 mm^2^. The mild steel (17.734.4) was integrated into the notch to fit exactly. The samples were oriented as shown in Figure 2.

The alternating immersion tests were performed using cycles of 30 min immersion and 150 min drying phases.

The salt spray test was performed according to DIN EN ISO 9227 standards. Galvanic-coupled samples were placed in the chamber at an angle of 30°. In the chamber, the pressure and temperature were maintained at 1 bar and 35 °C, respectively. After galvanic coupling evaluations, the corroded surfaces were cleaned ultrasonically with 50 ml/L phosphoric acid (H_3_PO_4_) solution for approximately 30 min at room temperature, rinsed with deionised water and ethanol. The surface appearance of specimens was monitored either via SEM-EDX or laser scanning confocal microscope (LSM 800, ZEISS) mentioned later.

All electrochemical measurements were repeated at least three times to verify the reproducibility of the results.

## 3. Results

### 3.1. Microstructural Analysis

According to the XRD diffractograms shown in Figure 3, the face-centred cubic (fcc) Al is the main phase (bold rhombus) in all alloys in both conditions. However, for Al-0.1In and Al-1Sn alloys, no indium- or tin-containing intermetallic phases were detected despite their relatively low solid solubility in aluminium matrix [31,32,37,38,39,40]. Only primary tetragonal In phases and tetragonal Sn phases were observed. Similarly in the Al-2Si alloy, large primary Si particles (fcc Si) were identified [41,42,43]. Since Ga and Zn possess relatively high solid solubility in aluminium, both elements were fully dissolved in the matrix without forming equilibrium phases [34,36]. For Al-3Zn-Ti-Mn-In alloys, only the fcc Al phase is visible and there is not any evidence of any phase formation of the five elements, Al, Zn, Ti, Mn and In, using XRD. After annealing treatment, tetragonal Sn, tetragonal In and fcc Si precipitations remained present.

Microstructures of the as-cast and solution-treated Al-based alloys are depicted in Figure 4. A typical dendrite microstructure is revealed in as-cast condition; additionally, Al-1Sn and Al-0.1In alloys displayed pure Sn/In precipitates, respectively, as shown in the higher magnifications’ micrographs. After solution treatment for 24 h, the as-cast microstructure was dissolved for almost all systems. In the case of Al, Al-0.2Ga, Al-0.5Cr and Al-5Zn, a precipitation-free microstructure could be achieved, while Al-1Sn, Al-2Si, Al-3Zn-Ti-Mn-In and Al-0.1In alloys still showed precipitations, with slightly increased concentrations of the alloying elements in the precipitates. EDX analysis was carried out for the solution-treated specimens. The composition of the intermetallic phases, labelled as (P), and matrix (M), are listed in Table 3. EDS analysis confirmed the information obtained by XRD.

### 3.2. Electrochemical Properties

#### 3.2.1. Open-Circuit Potential of Al-Based Alloys

Figure 5 shows the open-circuit potential (OCP) of the as-cast (a) and solution treated (b) Al-based alloys in 3.5 wt.% NaCl solution. After immersion in solution, the open-circuit potentials of most of the Al based-alloys reached a stable value in both conditions after a few minutes, except for the Al-0.1In alloy, which revealed fluctuations of potential without stabilisation during the whole test [21]. The average OCP measured values (E_op_) after 2 h are shown in Table 4.

In the as-cast condition, as well as for the commercial sacrificial anode (Al-3Zn-Ti-Mn-In), additions of Si, Zn and Cr in Al shifted the potential towards a positive direction. Meanwhile, additions of Ga and Sn can shift the potential to the more negative values of −1251 mV and −1443 mV vs. Ag/AgCl, respectively. However, after solution treatment, the Eop of most of the Al-based alloys became more negative, which is probably due to the dissolution of the as-cast microstructure. However, Al-Ga and Al-Sn showed a positive potential difference with respect to the as-cast condition of 62 and 9 mV, respectively.

#### 3.2.2. Potentiodynamic Polarisation Measurements

Potentiodynamic polarisation curves of as-cast (a) and solution-treated (b) Al-based alloys are depicted in Figure 6. The as-cast Al-1Sn, Al-0.1In, Al-5Zn and Al-3Zn-Ti-Mn-In alloys exhibited active corrosion behaviour in the anodic region, while Al-2Si, Al-0.5Cr, Al-0.2Ga and pure Al displayed lower current densities due to spontaneously passive behaviour. Apparently the solution treatment does not influence the corrosion behaviour of the alloys, since there were no remarkable differences in either the anodic polarisation or in the breakdown potential values (Table 4). The Al-1Sn, Al-3Zn-Ti-Mn-In, Al-0.1In and Al-5Zn alloys were still showing active behaviour in the anodic branch without passivation regions (Figure 6).

#### 3.2.3. Galvanic Coupling (ZRA)

This technique allows measurement of the current flow between the members of the galvanic couple [44], while simultaneously monitoring the mixed potential. This helps to select the material that can be used efficiently as sacrificial anodes. The mild steel (17.734.4) and the different aluminium base alloys form the galvanic couple. Mild steel acts as a cathode, while solution-treated Al-based alloys act as anodes. The solution treatment was carried out to retain the alloying elements in solid solution with the matrix, thus allowing the micro-galvanic corrosion between multiple intermetallic phases to be reduced [30,45]. The galvanic currents measured after 1 day and 21 days are shown in Figure 7 and Figure 8, respectively. During the first day, Al-Ga, Al-Cr, Al-Si, Al-Zn and pure Al systems gave the lowest galvanic currents, which were overlapping at average values of approximately 0.15 mA/cm^2^ (Figure 7). Al-3Zn-Ti-Mn-In provided a constant galvanic current of 0.23 mA/cm^2^ during the first day, while Al-In had an initial current density value of 1.45 mA/cm^2^, which decreased after 30 min and stabilised at an average current of 0.66 mA/cm^2^ with slight fluctuations. However, the Al-Sn alloy showed the highest galvanic currents; starting at 4.5 mA/cm^2^ from 3000 s, the density current reached an average value of 2.41 mA/cm^2^. Variations in galvanic current are known for their sacrificial anode applications, mainly due to the composition of the anode and the formation of corrosion products. For this reason, the current values are constantly monitored, so that structural failures can be prevented [46]. The active behaviour of Al-Sn was also observed in the anodic regions on the polarisation measurements (Figure 6). In terms of mixed potential, the Al-Si and Al-Cr systems reached values of −691 mV and−685 mV, respectively—close to Al with −730 mV—while the other systems achieved more negative values than Al. Al-Ga and the commercial anode showed a mixed potential value of −897 mV, Al-3Zn-Ti-Mn-In reached −910 mV and Al-Zn shifted the mixed potential to −962 mV. However, Al-In and Al-Sn alloys achieved the most-negative mixed potentials of −1135 mV and −1348 mV, respectively. Fluctuations in the potentials towards both more positive and more negative values can be explained by the breakdown and repair of the corresponding passive films [47]. For most of the materials, the galvanic currents and mixed potentials measured for the first day are consistent with the values estimated using the mixed potential theory [48] from the potentiodynamic polarisation curves (Table 5).

Figure 8 displays the performance of the Al-based alloys over 21 days. The time was limited due to some materials, such as Al-Sn, exhibiting extensive corrosion activity during the galvanic measurements. In some cases, the solutions became turbid, or on the mild steel surface, some red corrosion products were deposited. After 21 days, some Al-based alloys showed a reduction in the galvanic currents. The lowest galvanic currents were observed for Al-Cr with 0.05 mA/cm^2^ and Al-Zn with 0.07 mA/cm^2^. Al, Al-Si and commercial anode alloys reached similar values of 0.12 mA/cm^2^, 0.13 mA/cm^2^ and 0.14 mA/cm^2^, respectively. The performance as a sacrificial anode is related to the alloying additions and their distribution in the bulk of Al [49], thus promoting activation or passivation of the aluminium [50,51]. Al-In stabilised at 0.41 mA/cm^2^, while Al-Ga reached 0.54 mA/cm^2^. Despite the reduction of the galvanic current (0.90 mA/cm^2^), the Al-Sn alloy displayed the best performance in terms of sacrificial protection, with enormous self-degradation. The mixed potential curves showed similar trends to more positive potentials, shown by Al-Si at −686 mV, Al at-733 and Al Cr at −780. Relatively average mixed potentials were observed with Al-Zn at −966 mV and Al-Anode at −1071 mV, while Al-In, Al-Ga and Al-Sn exhibited the most-negative values at −1163 mV, −1277 mV and −1302 mV, respectively. According to NACE [17], a potential of −900 mV (Ag/AgCl) is required to cathodically protect steel in saline environments.

After measurements, the layer of corrosion product on the exposed area of the Al-based alloys was observed using SEM in backscattered modus (BSE) (Figure 9). It seems that the alloying selection could have influenced the corrosion-product layer formation on Al (Figure 9a). A mixed morphology of the corrosion-product layer was observed in Al-Cr, Al-Ga, Al-Si and Al-Zn-Ti-Mn-In (Figure 9b–d,g), which consists of a compact corrosion-product layer and a relatively porous region (marked with green and orange arrows, respectively). Al-Zn shows the mixed morphology mentioned above and an area that seems to be corrosion-product-free, outlined with the blue dashed line (Figure 9f). Al-Sn exhibits a thicker corrosion-product layer (Figure 9e), while Al-In depicts a remarkably porous layer (Figure 9h).

In order to obtain further information about the distribution of elements in the corrosion product film, elemental mappings were carried out on the layers, as shown in Figure 10. The corrosion product mainly consists of Al_2_O_3_/alloying element-oxides after galvanic coupling to steel in 3.5 wt.% NaCl. The porous regions described above are promoted due to the accumulation of the alloying particles underneath the corrosion-product layer, preventing the aluminium from forming an oxide layer as reported in [18,24,52]; this becomes visible by the lower distribution of O in that area from the element mapping (Figure 10). This phenomenon was most visible in the Al-Sn, Al-Si, Al-Zn, Al-Zn-Ti-Mn-In and Al-In alloys. A smaller influence of the alloying elements was observed in Al-Ga [53] and Al-Cr [25] alloys, possibly due to their lower additions in the Al matrix. Regarding the corrosion-product-free region highlighted with the blue dash line in Figure 9f, it was confirmed in Figure 10 that the O distribution does not match with the Al-rich region; similar regions were also observed in Al-Si mapping, and both corrosion-product-free regions were pointed out with the blue arrow in Figure 10. In the Al-In alloy, indium hinders corrosion product formation, promoting an extremely high number of pathways for chloride ions to reactivate Al.

Figure 11 exhibits the exposed areas after removal of the corrosion products. It could be seen that Si and Sn could prevent the formation of the corrosion product layer. A similar effect is promoted by In, Zn and the remaining alloying elements to a lesser extent (Figure 11).

Finally, the exposed areas of the steel after galvanic coupling were also visually evaluated using optical microscopy, and are shown in Figure 12. A full prevention of corrosion on the steel surface could not be achieved; however, the degree of protection of the steel can be ranked in descending order as follows: Al-0.2Ga > Al-0.1In > Al-1Sn > Al-5Zn > Al-3Zn-Ti-Mn-In > Al-2Si > Al > Al-0.5Cr.

### 3.3. Alternating Immersion and Salt Spray and Tests

Further corrosion investigations were carried out under aggressive conditions using alternating immersion and salt spray test in order to evaluate the sacrificial performance of the Al-based alloys. The galvanic-coupled assemblies depicted in Figure 2 were placed in the alternating immersion cell using 30 min immersion and 150 min drying phases. A second series of tests was carried out in an SST chamber according to the DIN EN ISO 9227 standard. The surface appearance of the specimens prior to both tests was achieved by a superficial inspection by comparison of photographs taken, as well as after 3 weeks and 6 weeks of exposure.

In the alternating immersion test (Figure 13), there were no significant visual differences in all systems between 3 weeks and 6 weeks of exposure compared with the before-test condition. Nevertheless, the Al-2Si, Al-1Sn and the pure Al alloys showed the best corrosion protection for the mild steel, since the formation of red corrosion products along the steel surface could not be noted.

Moreover, Al-1Sn also showed a large quantity of corrosion products; this is indication of strong sacrificial performance. Al-0.5Cr, Al-0.2Ga and Al-0.1In offered less protection for steel than that mentioned previously. However, Al-5Zn and Al-3Zn-Ti-Mn-In could not prevent rusting of the steel; in some cases, the red corrosion products started forming on the steel from the third week of exposure. From Figure 13 it can be observed that Al-0.1In also shows similar strong self-corrosion to the Al-1Sn system. The efficiency of sacrificial performance of the Al-based alloys can be listed in descending order as: Al-2Si > pure Al > Al-1Sn > Al-0.2Ga/Al-0.5Cr/Al-0.1In > Al3ZnTiIn > Al-5Zn.

The specimens exposed to SST formed a higher quantity of white corrosion products (Figure 14). In addition, Al-3Zn-Ti-Mn-In could not hinder corrosion after 3 weeks on the steel surface, while larger red corrosion products were observed on the steel for Al-0.2Ga and Al-0.1In. Al-0.5Cr, Al-2Si and pure Al displayed only slight rust formation. Contrary to the alternating immersion test, Al-5Zn depicted enhanced cathodic protection to steel, together with Al-1Sn. The cathodic protection performance of the Al-based anodes in SST could be ordered as follows: Al-5Zn > Al-1Sn > Al-0.2Ga > Al-0.1In > Al-0.5Cr > Al-2Si/Al > Al-3Zn-Ti-Mn-In.

After both tests, the Al, Al-5Zn, Al-3Zn-Ti-Mn-In samples (from the alternating immersion test) and Al-1Sn and Al-0.1In (from the salt spray test) were selected to remove the corrosion products and link the relationship between the preferential dissolution of the anodes and the formation of red rust on the steel. Confocal laser scanning images of the selected areas and their 3D images (in accordance with ISO 25,178 standards) are shown in Figure 15. Additionally, the volume loss was also determined using ConfoMap^®^ Surface Imaging and Analysis Software ST 7.2.7368 and reported in Table 6. Zn additions to the commercial anode promote a uniform degradation of the sacrificial anode, similar to that observed with pure Al; therefore, the contact between these anodes and steel is loosened, forming corrosion on the steel, thus resulting in the failure of its performance as a sacrificial anode (Figure 13). Similar behaviour was also observed with In additions (Figure 14) with a greater volume loss (Table 6). On the other hand, additions of activators such as Sn promote a non-homogeneous, network-like degradation that still allows direct contact between the anode and the steel, thus protecting the steel from corrosion (Figure 13 and Figure 14). However, the strong sacrificial behaviour of Al-Sn also generates larger consumption of the anode, as observed in its 3D image and confirmed by the volume-loss calculations in Table 6. This could be tailored by reducing the concentration of Sn in the aluminium matrix.

## 4. Discussion

In terms of OCP potential, noticeable variations occurred with the additions of In, Sn and Ga, as well as Cr to a lesser extent (Figure 5). The influence of indium in the aluminium matrix was considerably marked since the open circuit potential did not stabilise during the test. Muñoz et al. [22] found that indium could easily activate aluminium in the presence of chloride ions because indium promotes adsorption of chloride ions on the alloy surface, thus maintaining the aluminium matrix in an active state. Similarly, Sn has shown its ability as an activator for Al, shifting the potential of the binary Al-Sn alloy to more negative values compared to pure aluminium. Moreover, Ga [10] and Cr [54,55] are also added to aluminium in order to destabilise its oxide film and to promote the corrosion; however, in this study this property could not be observed and confirmed. Furthermore, Al-Ga and Al-Cr promote passivation and slow down dissolution in the anodic region. Less-extensive passivation was also noted for Al-Si. Despite Si, Zn and Al-3Zn-Ti-Mn-In shifting the potential to the nobler direction, these alloys showed active behaviours without any presence of passivation (Figure 6). Accordingly, an increase in the cathodic current density was observed for these systems. This increment of the cathodic activity might be the reason why no passivation region was visible. In fact, the cathodic currents were simply high enough to switch to the passive state and cause a breakdown in OCP. Additionally, the exposed area after galvanic coupling demonstrates the influence of the alloying elements on the corrosion product formation (Figure 9). Pure Al shows a dense and compact film with some cracks, but presents good adhesion (Figure 9a). Al-Cr, Al-Ga and Al-Si alloys (Figure 9b–d) depict a mixed morphology, which consists of compact film regions similar to those observed on Al, as well as porous sponge-like regions. It seems that additions of Ga and Cr promote passivation of Al since they have been retained in solid solution without formation of precipitates, which can serve as active microgalvanic cells (Figure 4) [25,26,27,54]. However, additions of Sn, Zn, Zn-Ti-Mn-In and In (Figure 9e–h), decrease the formation of the compact region, thus increasing the porous sponge-like areas that easily promote the adsorption of chloride ions on the alloy surface. Furthermore, element mapping shows the element distribution in the corrosion-product layer after galvanic coupling between the Al-based alloys and the mild steel (Figure 10). The greatest influence on the aluminium surface is observed with the Al-In alloy (Figure 9h); the bright oxygen areas seem to be related to In instead of Al. This was previously confirmed by elemental mapping after removing of the corrosion products (Figure 11). This feature renders indium very attractive as an alloying element for Al for marine applications [22,23]. For the special case of indium, a double effect of indium on the sacrificial aluminium anode’s surface is observed, first modifying the oxide layer on Al, then forming agglomerates of indium (see In distribution on mapping, Figure 10). It significantly favours the reactivity of aluminium, impeding its passivation, which is consistent with the literature [23,24]. Similarly, Sn promotes a similar effect. Keir et al. [29] attributed this behaviour to the creation of additional vacancies, which are occupied by Sn, thus improving cathodic protection properties. In all tests, the Al-Sn showed strong degradation as result of its exceptional performance as a sacrificial anode. On the other hand, Ga and Cr can also modify the aluminium oxide layer [53]; however, their activation performance to aluminium depends on their concentration and distribution in the alloy [21,53]; in this study, the additions were relatively lower. In order to evaluate the ability of Ga and Cr as activator elements for the aluminium alloys, it is recommended to increase their concentrations to values above 0.2% and 0.5% of Ga and Cr, respectively. Furthermore, corrosion-product-free regions observed in Al-Si and Al-5Zn (Figure 10) are a good example of activation of Al. This is a result of the higher tendency of Si to promote localised corrosion in Al-Si alloys, while Zn prevents corrosion product formation in Al [30,56,57]. This was later confirmed by elemental mapping after the removal of the corrosion products (Figure 11). In general, from the elemental mappings, the oxygen distribution (blue of varying intensity) indicates that the corrosion-product layer may consist of complexed Al_2_O_3_ together with a contribution of the alloying elements, as observed in almost all alloys, except Ga and Cr. Due to the results of Ma and Wen [49] it was expected that Al-3Zn-Ti-Mn-In alloy would possess the best performance as a sacrificial anode. They found that corrosion performance of Al-Zn-In-Mg-Ti-Mn is mainly activated by the cathodic Zn- and In-containing intermetallic phases at initial stages, generating pitting. Later on, the relatively homogenous corrosion is controlled by dissolution-precipitation of the In an Zn ions. However, the protective aluminium film could not be dissolved entirely, possibly because Zn and In were dissolved into the Al matrix after heat treatment. In addition to the influence of the alloying elements, intermetallic phase formation might take place; the performance of sacrificial anodes is influenced by the cathode/anode area ratio [13]. In practice, the C/A area ratio should be different from the unit [58]. This study evaluated the sacrificial performance of the binary systems in two conditions: when the cathode/anode area ratio is equal to one (ZRA measurements), and when the cathode/anode area ratio is less than one (alternating immersion and salt spray tests). For the ZRA measurements (Figure 7 and Figure 8), larger localised damages were observed after 21 days. According to the elemental mappings (Figure 10), all systems could modify the corrosion-product layer to some extent. However, the reactivation of aluminium is directly influenced by the diffusion of aggressive species, e.g., chlorides, through the various corrosion-product layers [21]. For systems with high self-corrosion such as Al-Sn and Al-In, the same cathode/anode area ratio increases the formation of the corrosion products (similar as in Figure 9b), generating excessive consumption of the anode. This condition does not guarantee a fully cathodic protection of steel, as observed in Figure 12.

On the other hand, for the alternating immersion and salt spray tests, although the area of the anode is much larger than that of the cathode, it did not prevent red rust formation on the steel notch, especially for the specimens containing Zn, In, Ga, Cr and the commercial anode (Figure 13 and Figure 14), while Sn exhibits an enhanced performance as a sacrificial anode, and to a lesser extent, silicon and aluminium. Vargel [59] indicated that the cathode/anode area ratio is not relevant, since galvanic corrosion always develops next to the zone of heterogeneous contacts. However, galvanic coupling results have shown that the cathodic protection of the Al-based alloys decreased after some time. The corrosion process is most probably caused by crevice corrosion [9], thus decreasing the cathodic protection performance. Nevertheless, laser scanning microscopy has shown that not all systems have the same degradation process; for example, Al-Sn depicts a network-like degradation, which is beneficial to establish and retain a direct contact between the anode and the steel, thus protecting the steel from corrosion.

## 5. Conclusions

Based on the presented results, some features should be considered when producing a sacrificial Al-based alloy anode for the cathodic protection of steel:

The alloying element additions should be tailored in order to shift the potential to suitable values where steel could be protected cathodically. However, when the potential is more negative than −900 mV, the sacrificial anode will suffer an unnecessary degradation.

A homogenous distribution of the alloying elements in the Al matrix is an important key to providing the conditions for perpetual-activation Al, avoiding its passivation.

High concentrations of Sn and In promote excessive self-corrosion, generating a faster degradation of the sacrificial anode. However, their higher galvanic currents could be useful to prevent steel from corrosion.

Thicker corrosion-product layers also decrease the efficiency of the Al-based alloys, due to a corrosion process that is controlled by diffusion of the aggressive species through the layer.

The corrosion evaluations lead to proposed additions of tin and indium as activators for Al-based alloys, due to their influence on hindering the formation of the natural passivation of Al.

Finally, the selection of suitable alloying elements and their required concentration would be crucial for the design of potential future Al-based alloy sacrificial coatings.

## Figures and Tables

**Figure 1 materials-15-01301-f001:**
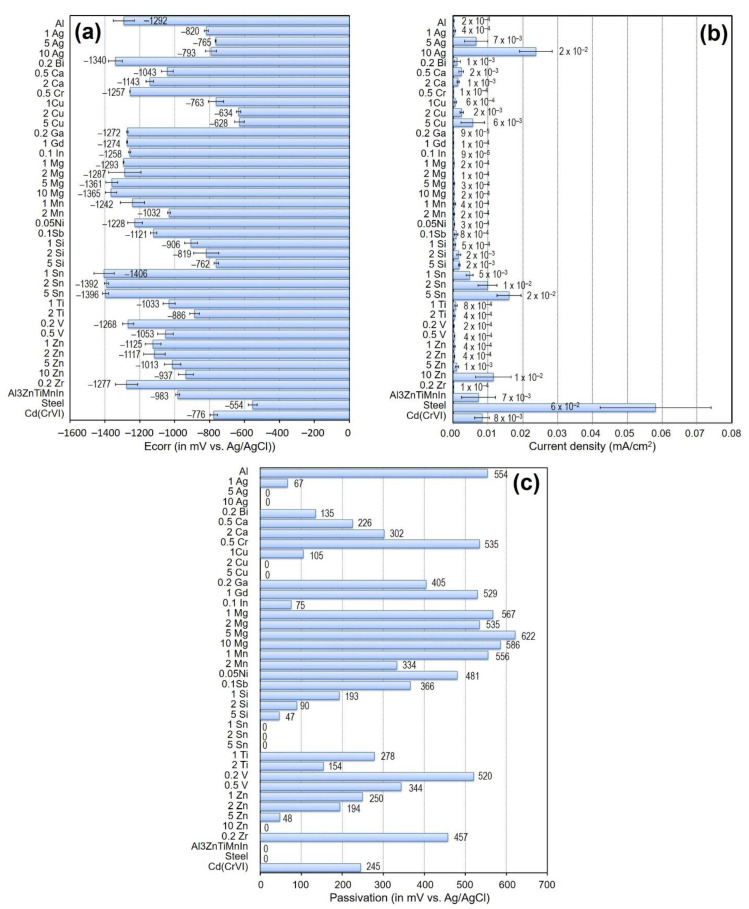
Corrosion properties of binary Al-based alloys measured in 3.5 wt.% NaCl: (**a**) corrosion potential (Ecorr), (**b**) corrosion current density and (**c**) passivation range. Additionally, mild steel, commercial sacrificial anode alloy (Al-3Zn-Ti-Mn-In) and a chromated Cd layer are included for comparison.

**Figure 2 materials-15-01301-f002:**
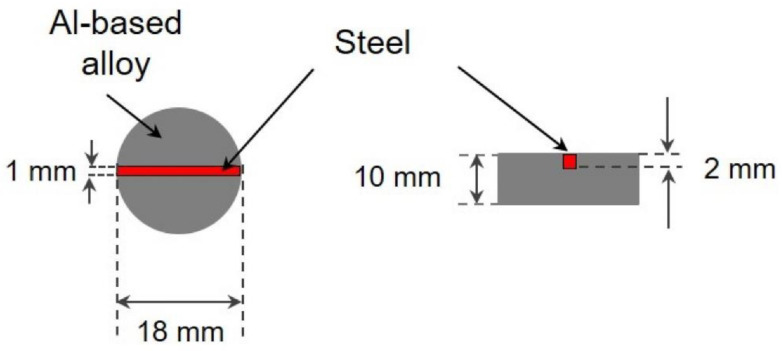
Galvanic coupled Al-based alloy–mild steel samples used for alternating immersion and salt spray tests.

**Figure 3 materials-15-01301-f003:**
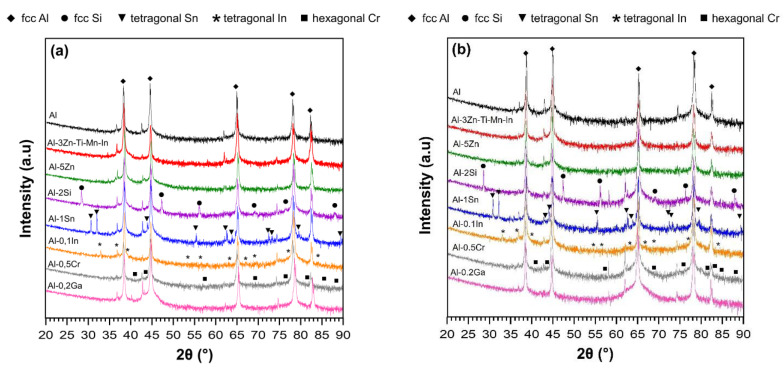
XRD spectra of Al-based alloys in: (**a**) as-cast and (**b**) 24 h solution treatment conditions, bold symbols indicate main phases according with the alloying element.

**Figure 4 materials-15-01301-f004:**
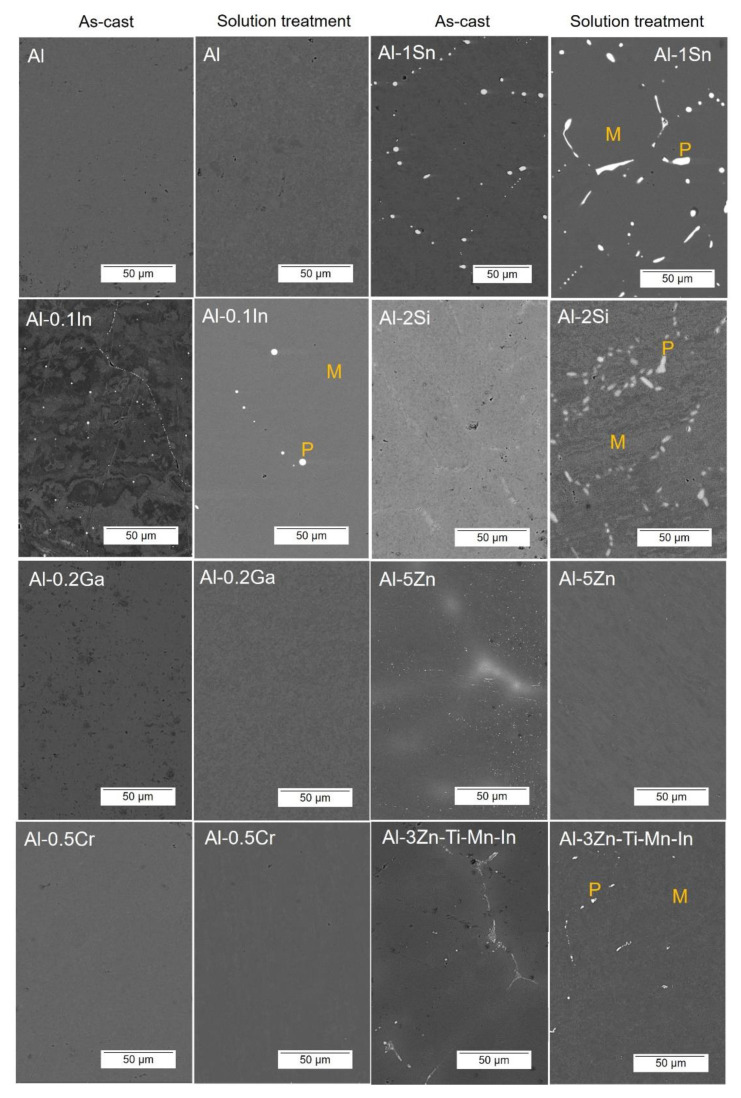
SEM images in the backscattered electron (BSE) mode of as-cast and solution treated Al-based alloys, EDS analysis was carried out in the intermetallic phases labelled as (P) and matrix (M).

**Figure 5 materials-15-01301-f005:**
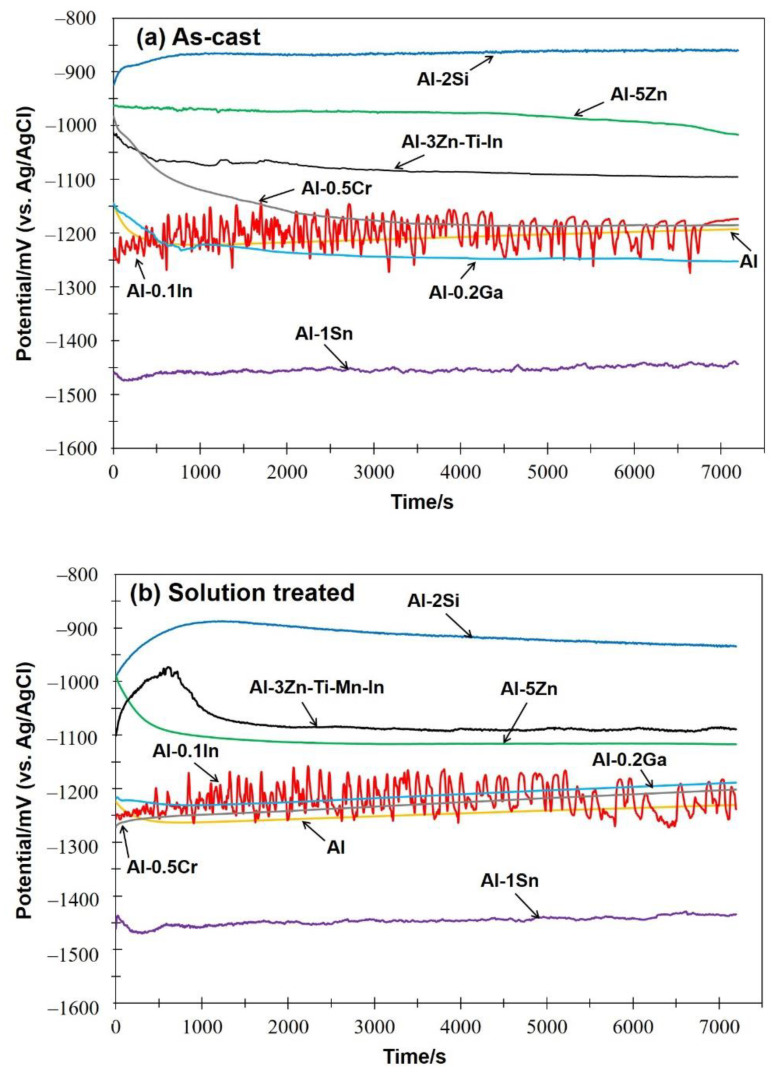
Open circuit potential (OCP) of as-cast (**a**) and solution-treated (**b**) Al-based alloys.

**Figure 6 materials-15-01301-f006:**
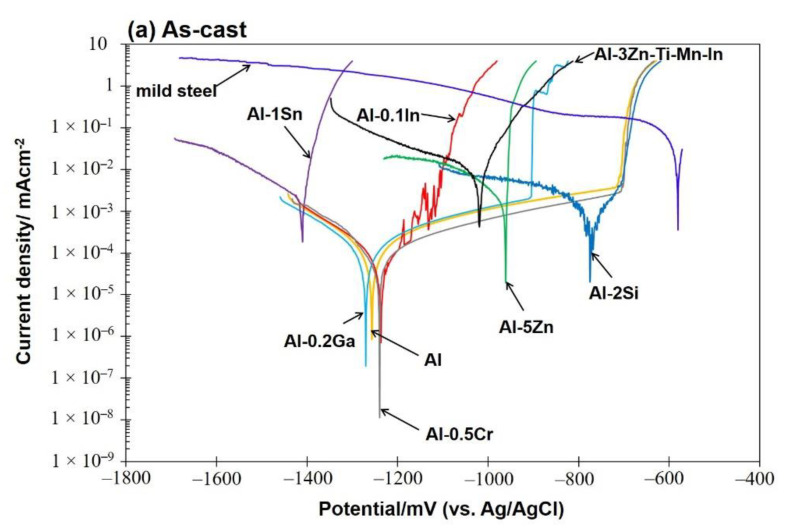
Potentiodynamic polarisation measurements of as-cast (**a**) and solution-treated (**b**) Al-based alloys and mild steel.

**Figure 7 materials-15-01301-f007:**
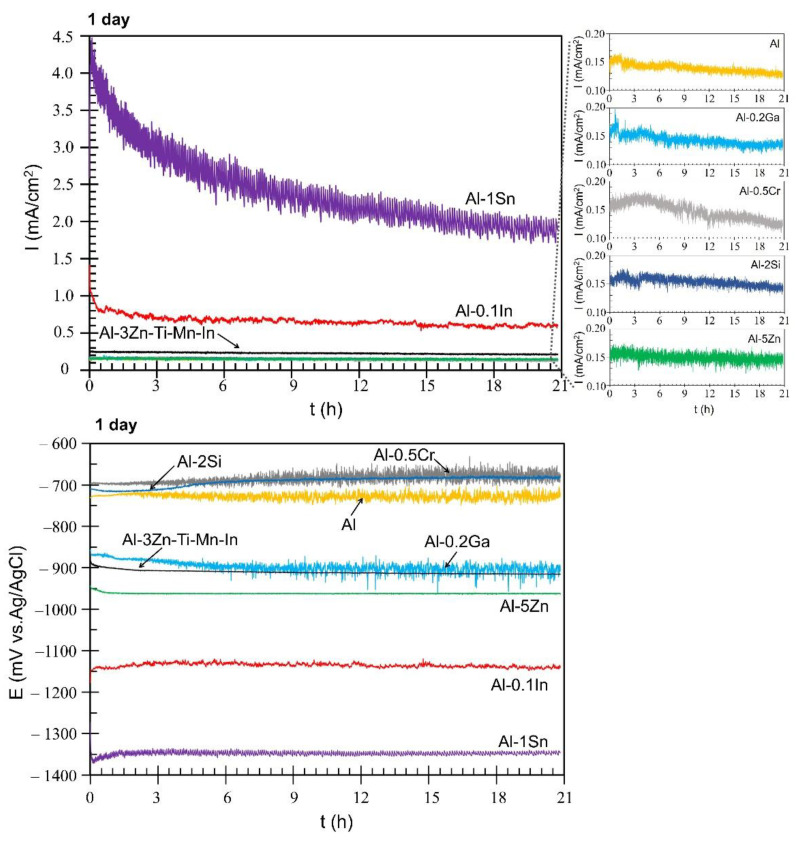
Galvanic current (I) and mixed potential (E) of the galvanic couple: mild steel and solution-treated Al-based alloys after 1 day.

**Figure 8 materials-15-01301-f008:**
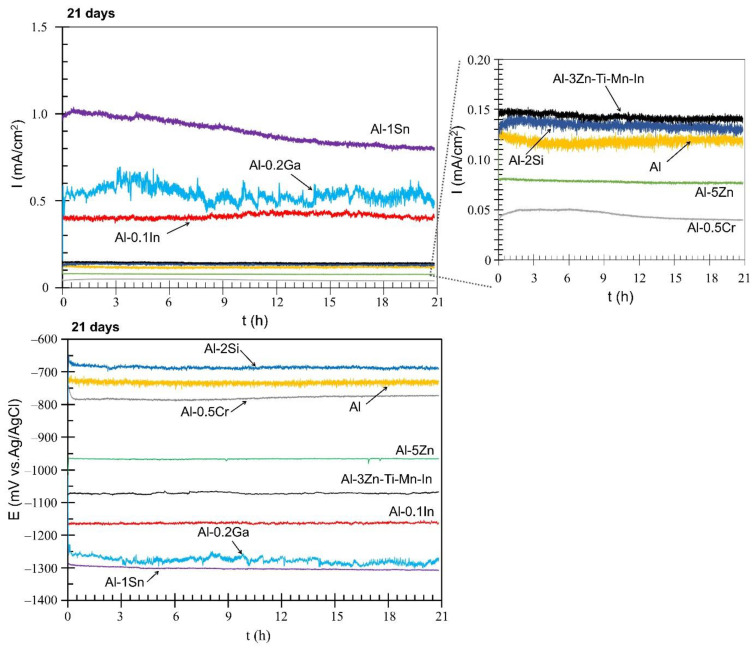
Galvanic current (I) and mixed potential (E) of the galvanic couple: mild steel and solution-treated Al-based alloys after 21 days.

**Figure 9 materials-15-01301-f009:**
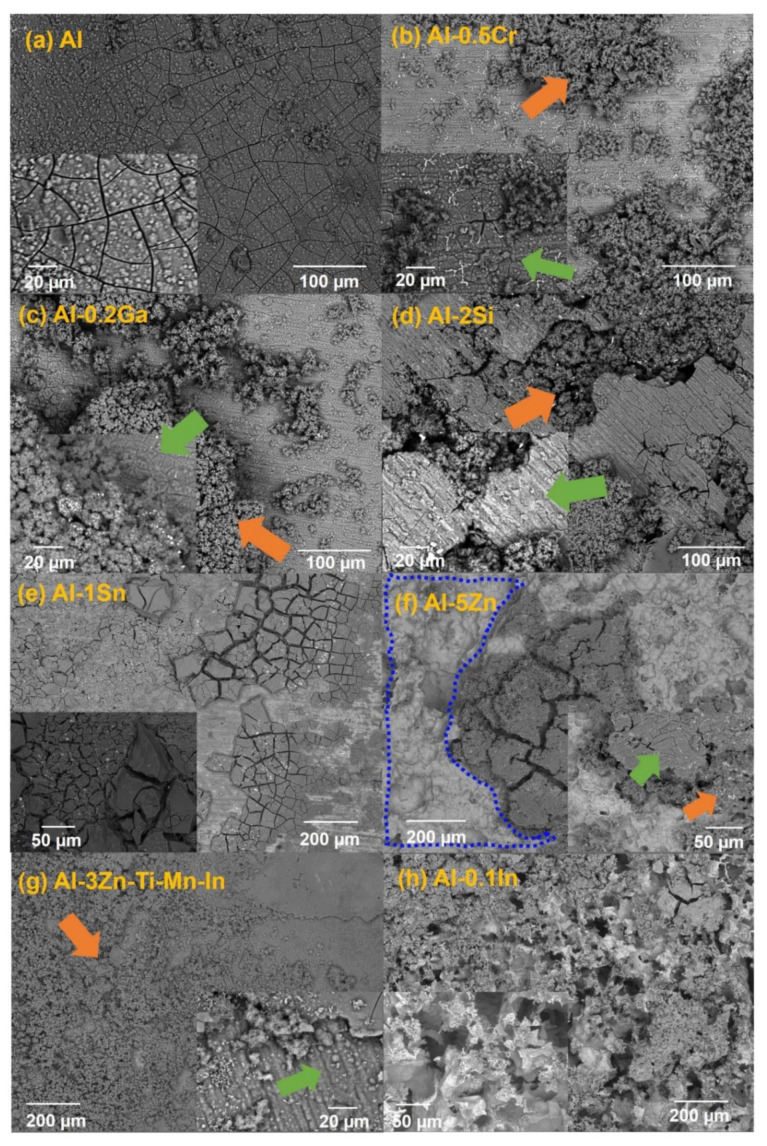
Corrosion product film formation after galvanic test (at high and low magnifications) of (**a**) Al, (**b**) Al-0.5Cr, (**c**) Al-0.2Ga, (**d**) Al-2Si, (**e**) Al-1Sn, (**f**) Al-5Zn, (**g**) Al-3Zn-Ti-Mn-In and (**h**) Al-0.1In, where green arrows represent a compact corrosion-product layer, while orange arrows show a porous corrosion-product layer and the blue dash line exhibits an corrosion-product-free region.

**Figure 10 materials-15-01301-f010:**
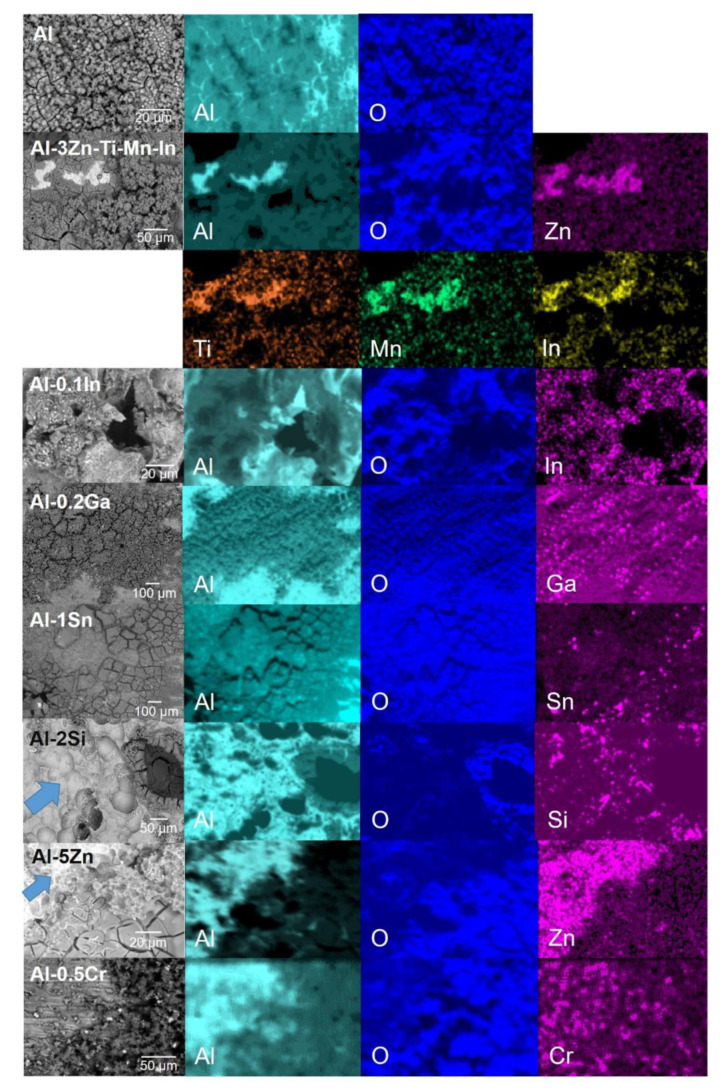
SEM-EDS elemental maps of the corrosion product film formation on the Al-based alloys after galvanic coupling with mild steel in 3.5 wt.% NaCl solution; blue arrows show corrosion-product-free regions.

**Figure 11 materials-15-01301-f011:**
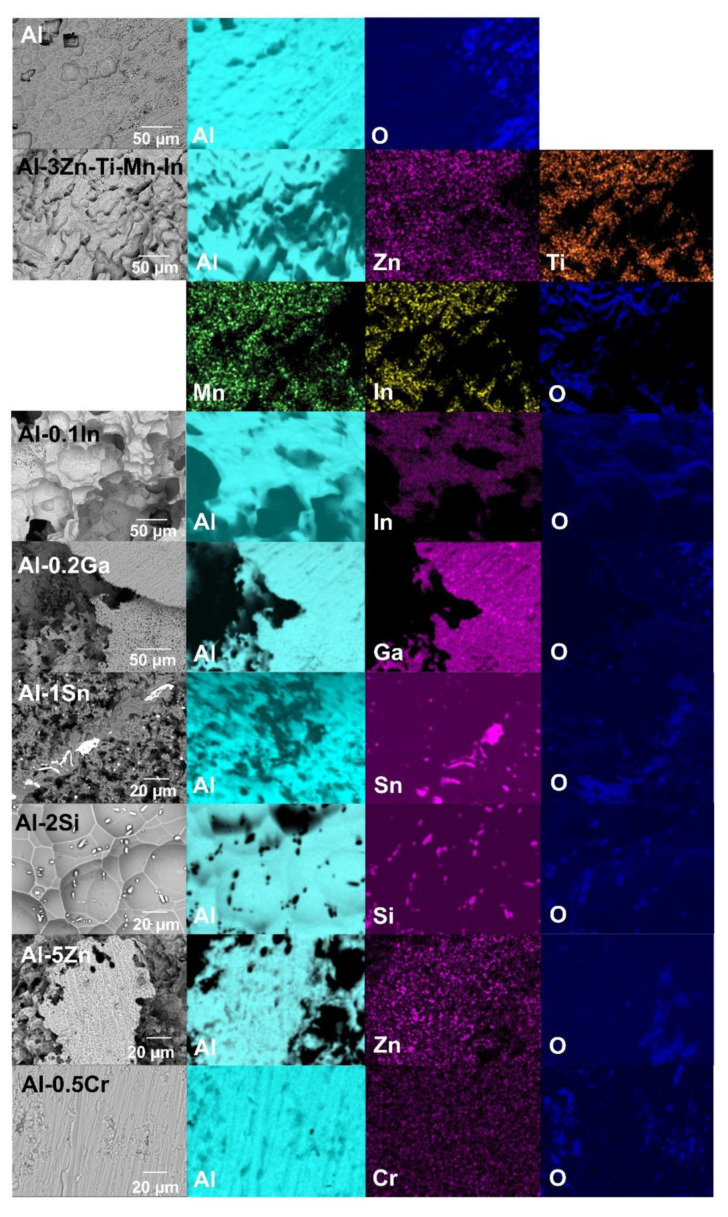
SEM-EDS elemental maps after removal of corrosion-product layer with phosphoric acid solution of the Al-based alloys after galvanic coupling with mild steel.

**Figure 12 materials-15-01301-f012:**
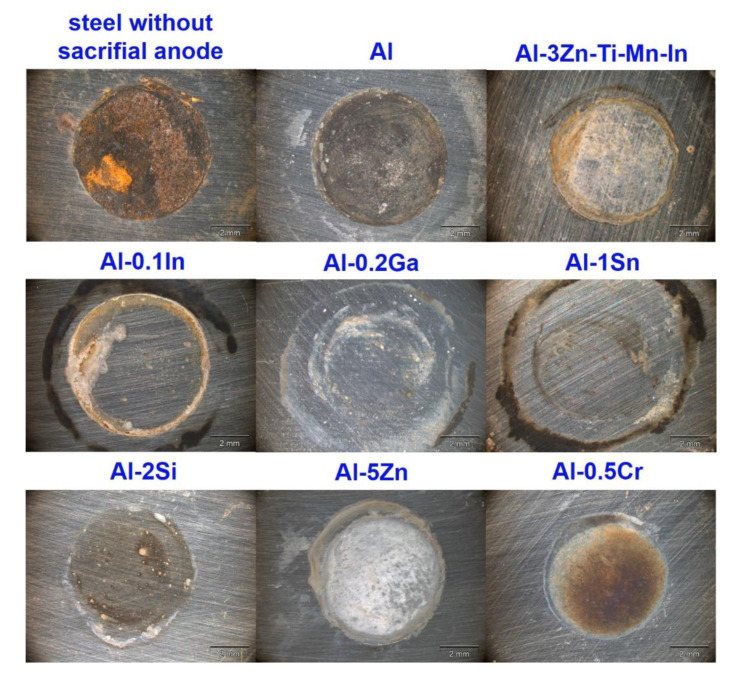
Optical images showing the protective effect for the steel sheet using the different sacrificial anodes after 21 days’ galvanic coupling.

**Figure 13 materials-15-01301-f013:**
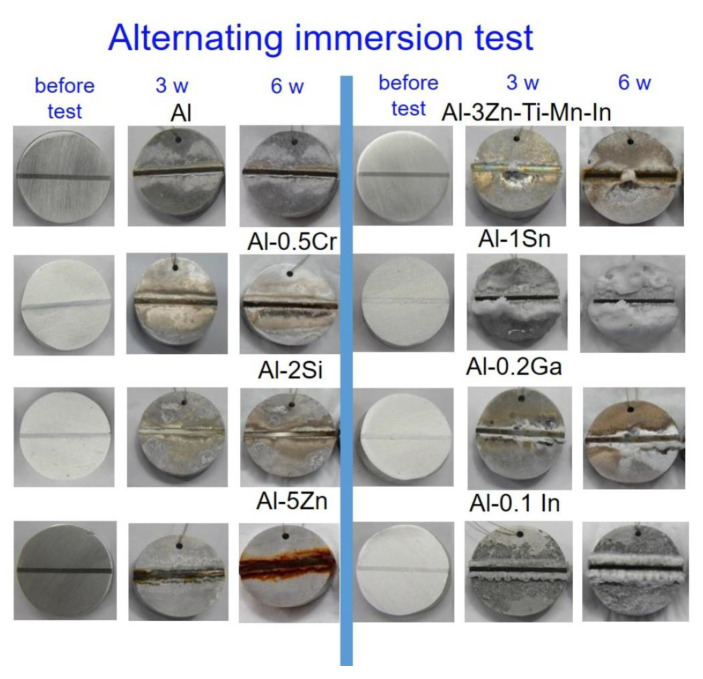
Images of the galvanic-coupled samples of mild-steel-Al-based alloys before testing, after 3 weeks and 6 weeks exposure in alternating immersion test.

**Figure 14 materials-15-01301-f014:**
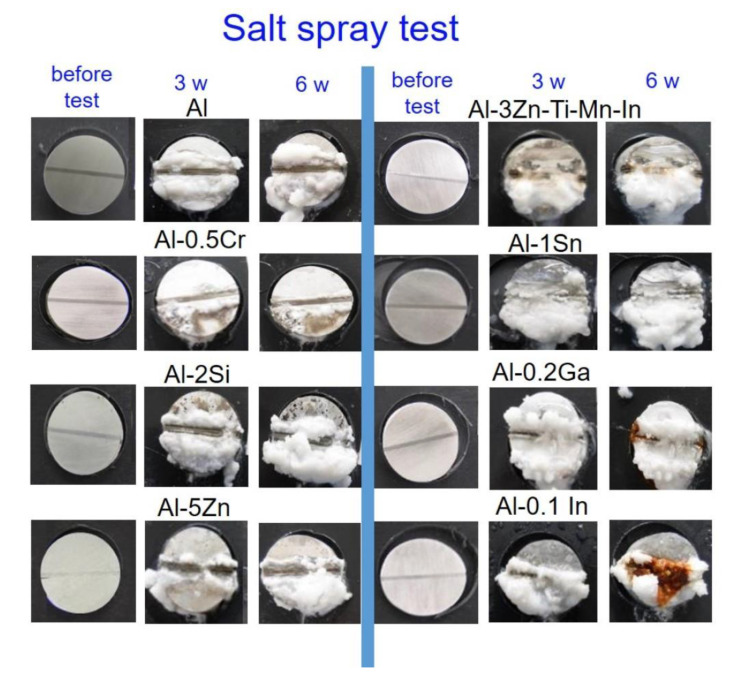
Images of the galvanic-coupled samples of mild-steel-Al-based alloys before testing, after 3 weeks and 6 weeks exposure in salt spray test.

**Figure 15 materials-15-01301-f015:**
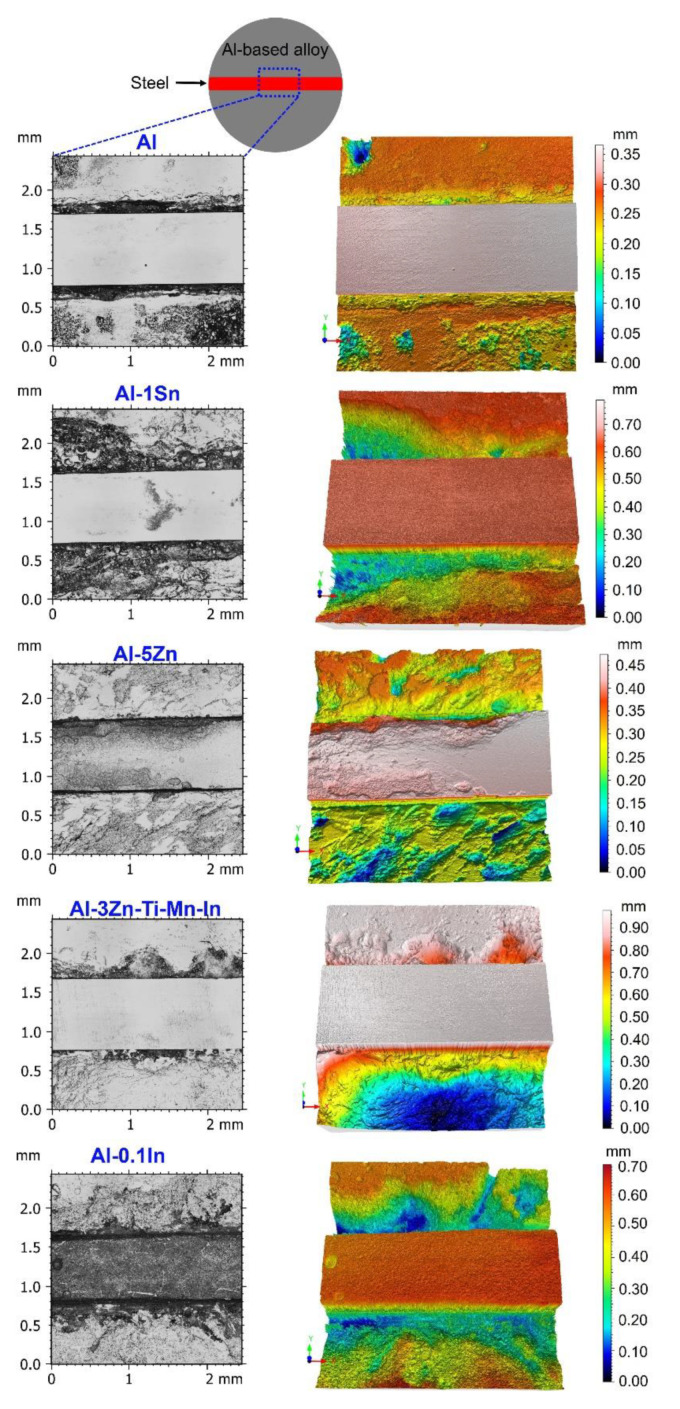
Selected confocal laser scanning images on selected area of the galvanic couple assembly and corresponding 3D images after corrosion product removal.

**Table 1 materials-15-01301-t001:** Solution treatment temperatures of binary Al-based alloys.

Alloy	Solution Annealing Temperatures (°C)
Al	600
Al-3Zn-Ti-Mn-In	530
Al-0.5Cr	600
Al-0.2Ga	600
Al-0.1In	600
Al-2Si	530
Al-1Sn	200
Al-5Zn	530

**Table 2 materials-15-01301-t002:** Chemical composition of binary Al-based alloys determined by X-ray fluorescence.

Alloy	Element								
	Zn (wt.%)	Ti (wt.%)	In (wt.%)	Mn (wt.%)	Cr (wt.%)	Ga (wt.%)	Si (wt.%)	Sn (wt.%)	Al (wt.%)
Al	0.04	0.01	<0.01	-	<0.01	-	0.18	<0.01	Bal. *
Al-3Zn-Ti-Mn-In	3.19	0.05	0.03	0.36	-	-	-	-	Bal.
Al-0.5Cr	-	-	-	-	0.56	-	0.17	-	Bal.
Al-0.2Ga	-	-	-	-	-	0.28	0.16	-	Bal.
Al-0.1In	-	-	0.21	-	-	-	0.20	-	Bal.
Al-2Si	-	-	-	-	-	-	2.72	-	Bal.
Al-1Sn	-	-	-	-	-	-	0.10	1.04	Bal.
Al-5Zn	5.23	-	-	-	-	-		-	Bal.

* Bal.—balance.

**Table 3 materials-15-01301-t003:** SEM-EDS analysis of solution-treated Al-based alloys.

Alloy/Point				Element					
	Sn (wt.%)	In (wt.%)	Zn (wt.%)	Mn (wt.%)	Ti (wt.%)	Si (wt.%)	Cr (wt.%)	Ga (wt.%)	Al (wt.%)
**Al-0.5Cr**	-	-	-	-	-	-	0.43 ± 0.06	-	Bal. *
**Al-0.2Ga**	-	-	-	-	-	-	-	0.34 ± 0.09	Bal.
**Al-0.1In**									
Precipitates (P)	-	16.12 ± 3	-	-	-	-	-	-	Bal.
Matrix (M)	-	0.15 ± 0.03	-	-	-	-	-	-	Bal.
**Al-1Sn**									
Precipitates (P)	94.89 ± 1	-	-	-	-	-	-	-	Bal.
Matrix (M)	0.13 ± 0.03	-	-	-	-	-	-	-	Bal.
**Al-2Si**									
Precipitates (P)	-	-	-	-	-	89.77 ± 4	-	-	Bal.
Matrix (M)	-	-	-	-	-	2.47 ± 0.03	-	-	Bal.
**Al-3Zn-Ti-Mn-In**									
Precipitates (P)	-	3.21 ± 0.26	5.96 ± 1.1	5.92 ± 3.6	1.27 ± 0.1	-	-	-	Bal.
Matrix (M)	-	2.50 ± 0.12	6.09 ±0.31	1.34 ± 0.04	1.04 ±0.03	-	-	-	Bal.
**Al-5Zn**	-	-	4.88 ± 0.02	-	-	-	-	-	Bal.

* Bal.—balance.

**Table 4 materials-15-01301-t004:** Corrosion properties of the Al-based alloys in as-cast and solution-treated conditions.

**As-Cast**	**E_op_ (mV)**	**E_corr_ (mV)**	**I_corr_ (µA/cm^2^)**	**E_break_ (mV)**
Al	−1193 ± 32	−1257 ± 33	0.11 ± 0.06	−722
Al-0.1In	−1173 ± 28	−1237 ± 27	0.08 ± 0.004	−1196
Al-0.2Ga	−1251 ± 29	−1270 ± 39	0.07 ± 0.02	−912
Al-0.5Cr	−1185 ± 32	−1238 ± 29	0.11 ± 0.02	−709
Al-1Sn	−1443 ± 8	−1410 ± 1	1.91 ± 0.47	-
Al-2Si	−860 ± 11	−775 ± 12	2.37 ± 0.65	−714
Al-3Zn-Ti-Mn-In	−1095 ± 3	−1020 ± 62	11.36 ± 0.6	-
Al-5Zn	−1017 ± 36	−961 ± 16	1.77 ± 1.34	-
**Solution-Treated**	**E_op_ (mV)**	**E_corr_ (mV)**	**I_corr_ (µA/cm^2^)**	**E_break_ (mV)**
Al	−1230 ± 36	−1287 ± 31	0.16 ± 0.05	−738
Al-0.1In	−1239 ± 19	−1260 ± 21	0.09 ± 0.006	−1193
Al-0.2Ga	−1189 ± 41	−1259 ± 36	0.07 ± 0.04	−874
Al-0.5Cr	−1202 ± 48	−1258 ± 61	0.16 ± 0.02	−718
Al-1Sn	−1434 ± 3	−1403 ± 6	4.52 ± 1.0	-
Al-2Si	−935 ± 65	−900 ± 68	2.37 ± 0.57	−730
Al-3Zn-Ti-Mn-In	−1091 ± 3	−964 ± 39	6.83 ± 3.44	-
Al-5Zn	−1118 ± 17	−994 ± 43	1.09 ± 0.48	-

**Table 5 materials-15-01301-t005:** Galvanic current and mixed potential (estimated and measured) of the solution-treated Al-based alloys vs. mild steel.

Solution-Treated Alloy	Estimation from Polarisation Curves		After 1 Day		After 21 Days	
	Galvanic Current (mA/cm^2^)	Mixed Potential (mV)	Galvanic Current (mA/cm^2^)	Mixed Potential (mV)	Galvanic Current (mA/cm^2^)	Mixed Potential (mV)
Al	0.17 ± 0.01	−710 ± 7	0.14	−728	0.12	−733
Al-0.1In	0.92 ± 0.02	−1081 ± 6	0.66	−1135	0.41	−1163
Al-0.2Ga	0.20 ± 0.01	−799 ± 19	0.14	−897	0.54	−1277
Al-0.5Cr	0.16 ± 0.01	−689 ± 2	0.15	−685	0.045	−780
Al-1Sn	2.32 ± 0.23	−1321 ± 12	2.41	−1348	0.90	−1302
Al-2Si	0.17 ± 0.02	−697 ± 6	0.15	−691	0.13	−686
Al-3Zn-Ti-Mn-In	0.29 ± 0.02	−899 ± 9	0.22	−911	0.14	−1071
Al-5Zn	0.36 ± 0.02	−931 ± 7	0.15	−962	0.07	−966

**Table 6 materials-15-01301-t006:** Volume-loss calculation of selected confocal laser scanning images after corrosion product removal.

Galvanic Couple	Volume Loss (µm^3^)
Steel-Al	4.5 × 10^7^
Steel-Al-1Sn	5.8 × 10^8^
Steel-Al-5Zn	1.1 × 10^8^
Steel-Al-3Zn-Ti-Mn-In	4.4 × 10^8^
Steel-Al-0.1In	6.8 × 10^8^

## Data Availability

Data sharing is available by request to the corresponding author, due to the data forming part of an ongoing study.

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
