# Peer review of "Cathodic Protection of Mild Steel Using Aluminium-Based Alloys"

_materials, 2022, doi:10.3390/ma15041301_

Round 1
Reviewer 1 Report
- Is bracket necessary after “SST” (page 3, line 102)?
- What are the samples dimensions in “2.2 Microstructure analysis” (page 6, lines 146-149)? Are these samples same as in “2.3 Electrochemical measurements”? How were samples cut out from the cast billets?
- It seems like it should be “range” in line 152 of page 6.
- All the scale bars at Figure 4 are 50 μm except one (Second line, first image from the left for Al-0.1In) – 20 μm. Is it correct? Not a mistake?
- You mention reference [21] in line 247 of page 10. It is given because there is explanation for fluctuations of potential for Al-0.1In alloy? Am I right? If it is possible, please, give very brief explanation for these fluctuations appearing in the text of your article.
- Page 22, lines 441-444 and Figure 15: how 3D images were created? Was some special software used? Please, describe it briefly.
Reviewer 2 Report
This study well demonstrated the corrosion effects of various Al-based alloys for steel. The results are considered to be beneficial to researchers aiming to develop Cd-free coatings. As listed below, several parts should be explained or discussed more in detail.
- How/Why do the authors select the composition of each alloy? The reason should be provided.
- How did the authors determine the solution annealing temperature for each alloy, as shown in Table 1?
- As listed in references, many researches seem to be published on the effects of additives on corrosion properties, so the results obtained in this study should be compared to those previously reported ones, in order to clarify the novelty of this work.
- The potential of the Al-based alloy coatings should be compared to the typical coatings including Cd and be discussed more in detail.
Reviewer 3 Report
Comments of materials-1547278
The main weaknesses of the manuscript:
- Your reference list is quite outdated, with more than half the papers more than 10 years old.
- How is the reproducibility of EDS? This could be for example, by including a mean and standard deviation in the Tables 3.
- How to analyze or calculate the methods in Figure 5 and Table 3? Please elaborate. So, are there relevant references to support it?

Reviewer 4 Report
The authors present an extensive collection of data that are of interest to a user in this field. Although the results are well presented, the less informed reader will have some difficulty following the content. The discussion of the results and the comparison with the literature clearly show the complexity of the topic.
The sample preparation and examination methods are well described. The same applies to the compilation of the measurement results. Tables and figures are clear and contribute to a better understanding.
In the conclusions, the authors suggest additions of tin and indium as activators for Al-based alloys. In this context, it would be helpful if the data for the galvanic couple steel – Al-0.1In could be added to Table 6 (volume loss) and Figure 15 (confocal laser scanning images).
Some typos have been identified and should be corrected.
Line 102: A bracket is missing after salt spray test.
Line 143: Replace x-ray with X-ray.
Lines 165 and 168: Here -1 must be written as an exponent.
Lines 214, 217, 218, 220: The term as-cast should be spelled consistently.
Line 249: Figure (b) partially covers Figure (a). The labeling of the x-axis is not completely legible.
Line 365: For sample (g) it has to be called Al-3Zn-Ti-Mn-In.
The following information should be added:
Which samples are shown in Figure 15, the samples from the alternating immersion test or those from the salt spray test? Is there an explanation why the both sides of the alloy Al-3Zn-Ti-Mn-In have such different material losses?
Reviewer 5 Report
This manuscript investigates the cathodic protection of mild steel using Al-based alloys as sacrificial anode material. The authors present several results, and their discussion seems appropriate. However, some important revisions are suggested. Please, find in what follows my specific comments.
- Introduction, lines 68-95 and Figure 1. It is not clear whether the authors are anticipating own results or they are describing literature findings. For instance, where does Figure 1 come from? Many data are reported therein but references are not reported while some of the alloys shown in the Figure were not investigated in this work. I suggest to deeply revised this part.
- Materials and Methods: Is Chinese Supplier the vendor of Gallium and Indium? It seems a generic name.
- Materials and Methods, line 122: it is reported “… the aluminum was alloyed and waited for… “ What is the meaning of “waited”?
- Materials and Methods: Source (vendor) of the Al-3Zn-Ti-Mn-In or, alternatively, the source of titanium used to fabricate this alloy should be reported.
- Materials and Methods, Table 1: Why the annealing temperature is not the same for the alloys?
- Figure 3: Several peaks appearing in the XRD patterns have been not assigned.
- Figure 4: Why have been different magnification reported for as-cast and solution treated alloy in the case of Al-0.1In system?
- Results, lines 278-282. The meaning of these sentences is not clear.
- Many typos appear in the text.
Round 2
Reviewer 1 Report
In my view, all the necessary corrections were made. Paper can be recommended for publishing.
Author Response
We thank the reviewer for his/her positive feedback
Reviewer 2 Report
Such information on the reason for deciding the composition, annealing temperature in the author reply comments should be provided in the text, citing the relevant references, briefly and shortly in several sentences for each reason.
Author Response
Point 1. Such information on the reason for deciding the composition, annealing temperature in the author reply comments should be provided in the text, citing the relevant references, briefly and shortly in several sentences for each reason.
Response 1. We thank the reviewer for his/her important remark. Requested information was added in lines 115-118 and lines134-142. References were also updated lines 662-667.
Reviewer 5 Report
The authors fully addressed all my comments. The manuscript can be accepted for pubblication.Author Response
We are grateful to the reviewer for his/her positive feedback.